# Lax Pairs for the Modified KdV Equation

Georgy I. Burde

Swiss Institute for Dryland Environmental and Energy Research, Jacob Blaustein Institutes for Desert Research, Sede-Boker Campus, Ben-Gurion University, Midreshet Ben-Gurion 84990, Israel; georg@bgu.ac.il

**Abstract:** Multi-parameter families of Lax pairs for the modified Korteweg-de Vries (mKdV) equation are defined by applying a direct method developed in the present study. The gauge transformations, converting the defined Lax pairs to some simpler forms, are found. The direct method and its possible applications to other types of evolution equations are discussed.

**Keywords:** integrable equations; Lax pairs; direct method

## 1. Introduction

Integrability is a wide subject that comprises many deep ideas and can be applied to very diverse physical systems. In the context of soliton theory, integrable systems of nonlinear partial differential equations (PDEs) have been actively studied [1–5]. As a matter of fact, the problem of solving nonlinear partial differential equations (PDE's) is twofold: (i) how to recognize an integrable PDE, and (ii) how to define its solutions. In the first aspect of the problem, there are several alternative definitions of integrability of nonlinear PDEs: existence of the Lax pair, existence of infinitely many generalized symmetries (or existence of recursion operators), multi-soliton solutions, Bi-hamiltonian structure (infinitely many conserved quantities), Bäcklund transformation, Painleve property, etc. (see, e.g., [5,6] for reviews). Each of those properties may be applied to a given equation as a test for integrability and, in the literature, there exist lists of integrable systems, i.e., systems which came through certain integrability tests.

On the other hand, there is a second aspect of the problem: defining solutions of nonlinear PDE's, which is a fairly complex problem. In general, the above mentioned characteristics of integrability can be exploited for finding solutions (see, e.g., [1–6]). However, the methods, based on those characteristics, as the rule, do not straightforwardly lead to the solutions and applying them for finding solutions requires ingenuity. (For example, the inverse scattering transform (IST), that is at the center of the solitons theory, in principle, provides solvability of the equation but the solution of the corresponding spectral problem can be particularly delicate.) A number of powerful methods of finding solutions of nonlinear PDEs, independent of the methods used for testing integrability, have been developed (an overview of such methods is presented in the Appendix A).

Different concepts of integrability are interwoven quite closely. In particular, the concept of Lax pair, a possibility for an equation to be realized as the compatibility condition of two linear eigenvalue equations, the Lax pair [7], appears in several approaches. The notion of Lax pair has played a key role in the development of soliton theory, and in many cases the identification of a corresponding Lax pair has been the first step to recognize the integrable character of important nonlinear PDEs.

However, this feature, the Lax pair existence, is not easy to determine a priori from the equation itself. Obtaining the Lax pair is a highly nontrivial operation and no systematic, general approach exists for this. The methods developed for the construction of a Lax pair (some of which are reviewed in [8]) involve nontrivial mathematical problems, making it

impossible to algorithmically determine whether a given equation admits a Lax pair or to determine, for which parameter values, a class of equations admits a Lax pair. Direct methods, which are based straight on the Lax pair definition, may provide more perspective. A direct method for identifying equations possessing Lax pairs has been developed and applied to some types of equations in [8]. In particular, the method has been applied to the modified Korteweg-de Vries equation (mKdV), which is one of the prototypical examples of integrable equations. The equation has the form

$$u_t + r_1 u^2 u_x + u_{3x} = 0 \tag{1}$$

where $r_1$ is an arbitrary parameter that can be changed by scaling and subscripts of the form "$nx$" denote derivatives of the order $n$ with respect to $x$. Applying the method of [8] to Equation (1) yields two different branches of the Lax pairs, one of which is a single Lax pair and the second represents a one-parameter ($\epsilon$) family of Lax pairs. A particular case of the second branch corresponding to $\epsilon = 0$ coincides with the known Lax pair for the mKdV equation while other Lax pairs, the first branch and those of the second branch corresponding to $\epsilon \neq 0$, seemed to be new. Those results imply that a variety of the Lax pairs for the mKdV equation exists.

    This paper reports the following results. First, by applying a direct method, that discards the requirement of scaling invariance imbedded into the framework of the method of [8] (see more details in Section 2), it is shown that the variety of the Lax pairs for the mKdV equation can be even more extended. Instead of a single Lax pair of the first branch of [8], the method of the present study yields a three-parameter family of Lax pairs while the Lax pair of [8] becomes a degenerated case of that family. The reason for the extension is that the method, which is free of the restriction of scaling invariance, allows terms of different scaling weights to appear in the Lax pair.

    Second, gauge transformations have been applied to separate from a variety of possible Lax pairs those that can be identified as 'fake'. At the early stages of the development of soliton theory, the opinion was widespread that the discovery of a Lax pair associated with a nonlinear evolution PDE implied that that PDE was integrable. However, after the observation of the fake (weak) Lax pairs phenomenon [9], the fact that a Lax pair can be associated to a nonlinear PDE cannot be considered as a proof of its integrability. There is no formal definition of which Lax pairs should be classified as weak, but it is implied that weak Lax pairs are useless for finding solutions by the IST and/or constructing conservation laws. It is found by applying proper gauge transformations that only one Lax pair (coinciding with the known one) from a variety of the defined Lax pairs can be useful for finding solutions of Equation (1). The three-parameter family of the Lax pairs of the first branch (including the first branch of [8] as a particular case) can be converted by a gauge transformation into the form which, upon using Equation (1) and its differential consequences, does not include $u$. Note, that in most cases reported in the literature, a Lax pair is useless if it is gauge equivalent to a matrix without a spectral parameter since the presence of the spectral parameter is crucial for finding solutions by the inverse scattering transform. The present (more rare) case is when a spectral parameter is non-removable but there exist gauge transformations allowing the removal of all dependent variables of the associated nonlinear system from the Lax pair. Further, it is found that, for the Lax pairs of the one-parameter ($\epsilon$) family of the second branch (and so for the second branch of [8] coinciding with it), there exists a gauge transformation eliminating the parameter $\epsilon$, which reduces it to the known Lax pair for the mKdV equation.

    This paper is organized, as follows. In Section 2 following the Introduction, the direct method is outlined and discussed. The Lax pairs for the mKdV equation, obtained by applying the method, are listed in Section 3. The issue of gauge equivalence and the gauge transformations, converting the defined Lax pairs to simpler forms, are discussed in Section 4. Some remarks on the results and possible applications of the method are furnished in Section 5. In the Appendix A, an overview of the methods of finding solutions of nonlinear PDEs, independent of the methods used for testing integrability, is presented.

## 2. Direct Method

The Lax pair in an operator form

$$L\psi + \lambda\psi = 0; \quad \frac{\partial \psi}{\partial t} + A\psi = 0 \tag{2}$$

admitted by a given PDE is determined from the condition of compatibility of (2) with the PDE. The condition can be written, in the form convenient for calculations, as

$$(Lf(x))_t = L(Af(x)) - A(Lf(x)) \tag{3}$$

where $f(x)$ is an auxiliary function. It should only hold on solutions of the original PDE.

In the procedure of the method, the operators $L$ and $A$ are sought as linear differential operators expressed in powers of $D_x$, as follows

$$L = D_x^m + U^{(1)}(u, u_x, u_{2x}, \ldots)D_x^{m-1} + \cdots + U^{(m)}(u, u_x, u_{2x}, \ldots)I, \tag{4}$$

$$A = Q^{(0)}D_x^n + Q^{(1)}(u, u_x, u_{2x}, \ldots)D_x^{n-1} + \cdots + Q^{(n)}(u, u_x, u_{2x}, \ldots)I \tag{5}$$

where $D_x$ is the total derivative operator for the space variable $x$, $I$ is the identity operator, $n$ is the order of the original PDE, $Q^{(0)}$ is a constant and it is assumed that the functions $U^{(i)}(u, u_x, u_{2x}, \ldots)$ and $Q^{(i)}(u, u_x, u_{2x}, \ldots)$ depend only on finitely many derivatives. The initial assumptions include choosing the order $m$ of the differential operator (4) and assigning the functional forms of $U^{(i)}(u, u_x, u_{2x}, \ldots)$ (making "ansatz") as differential polynomials in $u$ and its derivatives with the coefficients to be determined.

After that, the "ansatz" is specified, the procedure of the method becomes completely algorithmic. The functions $Q^{(i)}$ and the unknown coefficients of the differential polynomials $U^{(i)}$ are determined from the relation obtained by introducing (4) and (5) into the Lax Equation (3) expressing the condition of compatibility of the system (2) with the original PDE. The resulting expression is a linear differential polynomial in $f(x)$ and its derivatives and, in view of arbitrariness of $f(x)$, its coefficients dependent on $U^{(i)}$, $Q^{(i)}$ and their derivatives should vanish. This provides $n + 1$ relations that are considered as differential equations for the functions $Q^{(i)}$. Some of the equations can be solved even without specifying the forms of the functions $U^{(i)}$. The remaining (usually two) equations contain time derivatives of $U^{(i)}$ so that one needs to assign the forms of the differential polynomials for $U^{(i)}$ and use the original PDE and its differential consequences for eliminating terms with time derivatives of $u$. As the result, one has two equations for one function $Q^{(n)}$, which are compatible if the coefficients of a differential polynomial in $u$ and its derivatives obtained by eliminating $Q^{(n)}$ vanish. After solving the algebraic equations expressing this condition, a single differential equation for the last unknown function $Q^{(n)}$ remains. Solving this equation completes the derivation, but to obtain the result in quadratures, additional constraints on the coefficients need to be imposed.

A direct method for the Lax pairs calculations has been developed in [8]. The method of [8] differs from the present one in two aspects. First, the scaling symmetry condition is imposed which implies that all the terms in the operators $L$ and $A$ have uniform weight. (In the case of Equation (1), if the weight of the $x$–derivative is assumed to be equal to one, $W(\partial_x) = 1$, then $W(u) = 1$, $W(u_x) = 2$ and so on, and, correspondingly, $W(u_t) = 4$.) The second difference, partially related to the scaling symmetry requirement, is that not only the forms of the functions $U^{(i)}$ but also the forms of $Q^{(i)}$ are assigned. Using the scaling invariance condition simplifies the equations that determine the Lax pair and allows to formalize the procedure: the functional forms of both $U^{(i)}(u, u_x, u_{2x}, \ldots)$ and $Q^{(i)}(u, u_x, u_{2x}, \ldots)$ can be algorithmically assigned and a finite system of algebraic equations for the unknown coefficients arising at the final stage of the procedure (instead of a system of differential equations for $Q^{(i)}$ of the present method) can be algorithmically solved by the Gröbner basis methods. However, the scaling invariance condition reduces the generality of the method: the Lax pairs that include terms of different scaling weights are out of its scope.

In particular, this prevents application of the method of [8] to nonhomogeneous, mixed scaling weight, equations, since the Lax pairs with the terms of lower and higher weights originating respectively from the lower and higher scaling weight parts of the equation inevitably arise for such equations. In general, the Lax pairs for both homogeneous and non-homogeneous equations may include terms of different scaling weights, as a practice of applying the method of the present study shows. In most cases, the method yields not just single Lax pairs, but multi-parameter families of Lax pairs consisting of terms of different scaling weights. An example of such a multi-parameter family of Lax pairs yielded by applying the method to the mKdV Equation (1) is given in Section 3. Although, as it is shown in Section 4, the Lax pairs of that family should be treated as weak, the appearance of such terms is a sign that one should keep the functional forms for the operators $L$ and $A$ as general as possible in order to avoid missing some variants. In this respect, the procedure of the present method, in which the ansatzes for $U^{(i)}$ include terms of different scaling weights and the forms of $Q^{(i)}$ are not assigned but determined in the course of calculations, is preferable.

### 3. Lax Pairs for the mKdV Equation

For the mKdV Equation (1), two different families of the Lax pairs, with the operator $L$ of order 2, are available using the method of the present study. The first family is given by

$$L = D_x^2 + (\mu_2 + 2\mu_1 u)D_x + \left(\mu_1\mu_2 u + \mu_1^2 u^2 + \mu_1 u_x\right)I, \tag{6}$$

$$A = q_0 D_x^3 + 3q_0\left(\mu_1 u + \frac{\mu_2}{2}\right)D_x^2 + 3q_0\left(\mu_1^2 u^2 + \mu_1\mu_2 u + \frac{\mu_2^2}{8} + \mu_1 u_x\right)D_x$$

$$+\mu_1\left(\left(q_0\mu_1^2 - \frac{r_1}{3}\right)u^3 + \frac{3}{2}q_0\mu_1\mu_2 u^2 + \frac{3}{8}q_0\mu_2^2 u\right.$$

$$\left.+\left(\frac{3}{2}q_0\mu_2 + 3q_0\mu_1 u\right)u_x + (q_0 - 1)u_{2x}\right)I \tag{7}$$

where $\mu_1$, $\mu_2$ and $q_0$ are arbitrary constants ($\mu_1 \neq 0$). Note that the scaling weight of the terms multiplied by $\mu_2$ is different from that of other terms.

The second family is defined by

$$L = D_x^2 + 2\epsilon u D_x + \frac{1}{6}\left(\left(r_1 + 6\epsilon^2\right)u^2 + \left(6\epsilon \pm \sqrt{-6r_1}\right)u_x\right)I, \tag{8}$$

$$A = 4D_x^3 + 12\epsilon u D_x^2 + \left(\left(r_1 + 12\epsilon^2\right)u^2 + \left(12\epsilon \pm \sqrt{-6r_1}\right)u_x\right)D_x$$

$$+\left(\epsilon\left(\frac{2r_1}{3} + 4\epsilon^2\right)u^3 + \left(r_1 \pm \sqrt{-6r_1}\epsilon + 12\epsilon^2\right)uu_x + \left(3\epsilon \pm \frac{1}{2}\sqrt{-6r_1}\right)u_{2x}\right)I \quad (9)$$

where $\epsilon$ is an arbitrary constant.

Two branches of Lax pairs have been identified for the mKdV equation in [8]. Their 'first branch' is a degenerate case ($q_0 = 0$, $\mu_2 = 0$, $\mu_1 = 1$) of the family defined by ((6), (7)) and their 'second branch' is ((8), (9)) (the notation coincides except for that their $\alpha$ should be replaced by $r_1$ and the sign of the operator $A$ should be changed).

### 4. Gauge Equivalence

For Lax pairs in matrix form (zero-curvature representation [2–4,10,11]), scalar Equation (2) is replaced by the matrix equations

$$D_x\Psi = X\Psi, \quad D_t\Psi = T\Psi \tag{10}$$

where $\Psi$ is a vector function on the jet space of $u$ and $X(\lambda, u, u_x, \ldots)$ and $T(\lambda, u, u_x, \ldots)$ are, in general, $(n \times n)$ matrices dependent on the spectral parameter $\lambda$. The Lax Equation (3) is replaced by the equation expressing the compatibility condition for (10), as follows

$$(D_t X - D_x T + [X, T])\Psi = 0 \tag{11}$$

where $[X, T] = XT - TX$ is the matrix commutator. Equation (11) is called the matrix form Lax equation or zero-curvature condition.

For any pair of matrices $(X, T)$ that satisfy (11), an infinite number of equivalent pairs

$$D_x \tilde{\Psi} = \tilde{X} \tilde{\Psi}, \quad D_t \tilde{\Psi} = \tilde{T} \tilde{\Psi} \tag{12}$$

may be found through a gauge transformation of the form

$$\tilde{\Psi} = G\Psi, \quad \tilde{X} = GXG^{-1} + D_x(G)G^{-1}, \quad \tilde{T} = GTG^{-1} + D_t(G)G^{-1} \tag{13}$$

where $G$ is a nonsingular matrix.

The gauge transformations that convert the families of Lax pairs ((6), (7)) and ((8), (9)) into some equivalent forms can be found. Let us consider first the one-parameter ($\epsilon$) family ((8), (9)). In what follows, it is set $r_1 = -6$ to simplify the formulas. A family of matrices $(X, T)$ corresponding to the operators ((8), (9)) is given by

$$X = \begin{pmatrix} 0 & 1 \\ -\lambda + (1 - \epsilon^2)u^2 - (1 + \epsilon)u_x & -2\epsilon u \end{pmatrix} \tag{14}$$

$$T = \begin{pmatrix} 4\epsilon\lambda u - 2(1 + \epsilon)uu_x + (1 + \epsilon)u_{2x} & 4\lambda + 2u^2 - 2u_x \\ \begin{matrix} -4\lambda^2 + (2 - 4\epsilon^2)\lambda u^2 + (2 - 2\epsilon^2)u^4 - 2\lambda u_x \\ -2(2 + \epsilon - \epsilon^2)u^2 u_x - 2(1 + \epsilon)uu_{2x} + (1 + \epsilon)u_{3x} \end{matrix} & \begin{matrix} -4\epsilon\lambda u - 4\epsilon u^3 \\ +2(1 + \epsilon)uu_x - (1 - \epsilon)u_{2x} \end{matrix} \end{pmatrix} \tag{15}$$

Applying the gauge transformation (13) with the matrix

$$G = \begin{pmatrix} e^{\epsilon \int u(x,t)dx} & 0 \\ \epsilon u(x,t) \, e^{\epsilon \int u(x,t)dx} & e^{\epsilon \int u(x,t)dx} \end{pmatrix} \tag{16}$$

to the family of matrices (14) and (15) yields the matrices $(\tilde{X}, \tilde{T})$ not containing the parameter $\epsilon$, as follows

$$X_1 = \begin{pmatrix} 0 & 1 \\ -\lambda + u^2 - u_x & 0 \end{pmatrix} \tag{17}$$

$$T_1 = \begin{pmatrix} -2uu_x + u_{2x} & 4\lambda + 2u^2 - 2u_x \\ -4\lambda^2 + 2\lambda u^2 + 2u^4 - 2\lambda u_x - 4u^2 u_x - 2uu_{2x} + u_{3x} & 2uu_x - u_{2x} \end{pmatrix} \tag{18}$$

The corresponding Lax pairs in an operator form are ((8), (9)) taken for $\epsilon = 0$. The Lax pair ((17), (18)) is the known Lax pair for the mKdV equation, which can be obtained from the Lax pair

$$X_{KdV} = \begin{pmatrix} 0 & 1 \\ -\lambda - v & 0 \end{pmatrix} \tag{19}$$

$$T_{KdV} = \begin{pmatrix} v_x & 4\lambda - 2v \\ -4\lambda^2 - 2\lambda v + 2v^2 + v_{2x} & -v_x \end{pmatrix} \tag{20}$$

for the KdV equation

$$v_t + 6vv_x + v_{3x} = 0 \tag{21}$$

through Miura's transformation

$$v = u_x - u^2 \tag{22}$$

The Lax pair ((17), (18)) is equivalent to another known Lax pair for the mKdV equation [3,12]

$$X_2 = \begin{pmatrix} -ik & -u \\ -u & ik \end{pmatrix} \tag{23}$$

$$T_2 = \begin{pmatrix} -4ik^3 - 2iku^2 & -4k^2u - 2u^3 - 2iku_x + u_{2x} \\ -4k^2u - 2u^3 + 2iku_x + u_{2x} & 4ik^3 + 2iku^2 \end{pmatrix} \tag{24}$$

to which it is related by the gauge transformation

$$G = \begin{pmatrix} 6(ik - u) & -6 \\ 6(ik + u) & 6 \end{pmatrix} \tag{25}$$

Analyzing the matrix forms of the first family of Lax pairs ((6), (7)) reveals that it can be converted by a gauge transformation into the form, which, upon using Equation (1) and its differential consequences, does not include $u$. In order not to overload the presentation, the matrix form of the Lax pair ((6), (7)) is not shown, only the transformation matrix is given below. The Lax pairs ((6), (7)), being written in matrix forms, can be reduced by the gauge transformation (13) with the matrix

$$G = \begin{pmatrix} e^{\mu_1 \int u(x,t)dx} & 0 \\ \mu_1 u(x,t) \, e^{\mu_1 \int u(x,t)dx} & e^{\mu_1 \int u(x,t)dx} \end{pmatrix} \tag{26}$$

to the constant matrices

$$\tilde{X} = \begin{pmatrix} 0 & 1 \\ -\lambda & -\mu_2 \end{pmatrix}, \qquad \tilde{T} = \begin{pmatrix} \frac{1}{2}\mu_2 q_0 \lambda & \frac{1}{8}q_0(8\lambda + \mu_2^2) \\ -\frac{1}{8}q_0\lambda(8\lambda + \mu_2^2) & -\frac{1}{8}q_0\mu_2(4\lambda + \mu_2^2) \end{pmatrix} \tag{27}$$

Such a Lax pair not containing the dependent variable $u$ is useless for finding solutions by the IST and/or constructing conservation laws. Thus, the Lax pairs ((6), (7)) should be treated as weak.

## 5. Concluding Comments

In the present paper, the results of application of the direct method to the mKdV equation are considered. It is demonstrated that the method can produce multi-parameter families of the Lax pairs. At the same time, by applying proper gauge transformations it is found that, (at least) in the case of that specific equation, the defined Lax pairs can be either reduced to a single Lax pair, which is known, or converted into the form not containing the dependent variable, which is useless for applications. Thus, only the known Lax pair is what remains from the variety of Lax pairs defined (and so from its particular case found in [8]).

As a matter of fact, the direct method should be most useful for the problem of classification of integrable equations of some specific type, but not as applied to a single equation. Being applied to an equation or system with parameters, the method yields conditions on the parameters for the Lax pair existence. Given that fake Lax pairs cannot be avoided, a positive result in testing a PDE for the existence of a Lax pair does not warrant placing the equation on the list of integrable equations. At the same time, a reliable negative result of testing may be considered as a strong argument against its integrability since it is commonly believed that a completely integrable nonlinear PDE can be associated with a Lax pair. In the context of the problem of classification of integrable equations, detecting equations that cannot be integrable should be as important as finding candidates for integrable equations. It allows for a substantial reduction in the list of candidates for integrable equations of that specific type. From a somewhat different perspective, it may enable complete classification if applying the method yields only equations that have been proved to be integrable.

The method developed in the present study provides a reliable test for Lax pairs of a quite general form. Practically, the freedoms in the choice of the "ansatz" for the differential operator *L* do not result in a loss of generality. With the capabilities of computer algebra, there are no principal obstacles to implementing calculations for any reasonable candidate. Note in this connection that the Lax pairs presented in this paper have been separated from substantially more complicated initial forms. With such general ansatzes, a negative result of testing a PDE with respect to the Lax pairs existence is as close to a proof as is possible using direct methods. Due to the fact that the requirement of scaling invariance is not imposed, the method can be effective in application to both homogeneous and mixed scaling weight equations.

**Funding:** This research received no external funding.

**Data Availability Statement:** Data is contained within the article.

**Acknowledgments:** I am grateful to the Reviewers for encouraging comments and suggestions for improvement of the presentation.

**Conflicts of Interest:** The authors declare no conflicts of interest.

### Appendix A. Methods of Defining Solutions of Nonlinear PDEs Independent of the Methods for Testing Integrability

The methods can be classified as belonging to two groups: (a) Symmetry (or symmetry-based) methods and (b) Direct methods. In the first group, the classical Lie-group method of infinitesimal transformations takes centre stage. In that method, the infinitesimal group generators are defined as solutions of 'determining equations' obtained from the condition of invariance of a PDE under the group transformations (see, for example, [13–15]). This so-called 'classical' method enables the reduction of PDEs to ordinary differential equations (ODEs), defining group-invariant solutions and finding transformations of the variables of a PDE, by which new solutions can be generated from known ones. (Also numerical methods, that make use of the symmetry in partial differential equations, have been developed as, for example, the Lie-group shooting method for solving Stefan problems [16], see also [17].)

Bluman and Cole [18] proposed a generalization of Lie's method for finding group invariant solutions, which they named the 'nonclassical' method. In this approach, the condition for the invariance of the PDE is replaced by weaker conditions for the invariance of the combined system of differential equations consisting of the original differential equation along with the 'invariant surface condition'. The set of solutions, potentially available with the help of this method, is larger than the set obtained by the classical method. However, unlike the situation for 'classical' symmetries, the system of determining equations is, in general, nonlinear and so the procedure of defining solutions with the nonclassical method is not completely algorithmic. In particular, Bluman and Cole, applying their method to the linear heat equation in [18], did not succeed in solving determining equations. For that reason, the method did not attract much attention and practically had never been used till 1989 when it was established by Levi and Winternitz [19] that the solutions of the Boussinesq equation, found by applying the direct method of Clarkson and Kruskal [20], can be recovered as invariant solutions under the non-classical symmetry groups admitted by the equation. This showed that, even though the nonclassical method appeared to be ineffective in application to the linear heat equation, its application to nonlinear equations may be much more fruitful. After that studies applying the nonclassical methods to nonlinear wave equations have flourished. The renewed interest in the nonclassical method resulted in both nonclassical and classical methods being used to generate many new symmetry reductions for several physically significant equations (see reviews in [21,22]). Nevertheless, because of the nonlinearity of the system of determining equations, nonclassical symmetries of many PDEs in physics and mechanics have not been found. The ways of simplifying the solution of the system of nonlinear determining equations of the nonclassical method, based on the Groebner basis method and Wu's method, are investigated in [23–25].

The ideology of the nonclassical method gave rise to other efficient and elegant methods of treating differential equations, which can be considered as generalizations and extensions of the classical Lie group method and of the nonclassical method, such as the weak symmetry method, the side condition method, the nonlocal symmetry method, the iteration of the nonclassical method, partially nonclassical method, the conditional Lie-Bäcklund symmetry, the differential constraint method and so on (e.g., [26–37], see also [38]). In many cases, the methods are interrelated.

Considering the methods belonging to the group (b), direct methods, one should start from the oldest, and, in a sense, classical, method of the solitons theory, Hirota's method [39]. The idea was to make a transformation of the equations into new variables, in which the equations turned out to be quadratic in the dependent variables and all derivatives appeared as Hirota's bilinear derivatives (this is called "Hirota bilinear form"), and then multisoliton solutions appear in a simple form of exponents with the travelling wave argument. It is appropriate here to refer to the review of applications of Hirota's method for equations in the Korteweg-de Vries class in [40] where, in particular, use of the existence of multisoliton solutions as an integrability condition is discussed. Note, however, that application of Hirota's method to a specific nonlinear equation requires some ingenuity since constructing the bilinear form of the equation is not algorithmic. In [41], a simplified form of the Hirota method, which circumvents the stage of creating the bilinear form, has been created. The significant modification of Hirota's method, applicable both to the original and simplified versions, has been developed in [42]. In the method of [42], the constant coefficients of exponents in the solution form are replaced by polinomial functions of independent variables, which enables the algorithmic construction of solutions describing interactions of different types of solitons. In particular, solutions describing interactions of 'static' solitons with moving ones, are constructed in [42].

Reviewing other currently used direct methods, one has to start from the direct method of Clarkson and Kruskal (CK) [20]. The basic idea is to seek the solution of an equation under consideration in the form that could be considered as the most general form for similarity solutions (see [43]). As a matter of fact, all the direct methods, developed after the CK method, are based on the ideology of the CK method in a sense that their starting point is a quite general ansatz for solutions, which are used for reducing the problems of solving PDEs to solving ODEs. The difference with the CK method may be either in the form of the ansatz or in the way of using it (for example, in the direct method of [44], the PDE is reduced to an overdetermined system of ODEs). Next, establishing the relation between the CK method and the nonclassical method in [19] resulted in the focus of much of the research becoming finding the equivalent symmetry-based formulation for a specific direst method (and vice-versa). The current state of the art is, in general, separating the methods into the groups of symmetry-based methods and direct methods is somewhat conventional. Practically, for all direct methods, stemming from the ideology of the CK method, a symmetry-based formulation can be found (e.g., all the solutions, obtained by a direct method in [44], can be recovered as invariant solutions of the symmetry-based method of [33]). Reviews of relationships between the methods, developed as generalizations and extensions of classical or nonclassical methods, and various direct methods can be found in [45–47]. Probably, the only exception from the above statement about the possibility of finding a symmetry-based reformulation for a given direct method is the method developed in [48] and further generalized in [42]. The particular feature of that method resides in using the "potential", space integral of the dependent variable, as the independent variable, which is a suitable tool for identifying the solitary wave solutions. The form of solutions in the original variables is not prescribed and even cannot be defined in advance since the potential as a function of the original variables can be derived only upon finding the solution. Therefore, the method does not belong to the variety of methods that, like the CK method, start from the ansatz in the original variables and that feature makes it impossible to find the equivalent symmetry-based formulation for the method of [48].

Finally, such methods, as the 'sine–cosine method', the 'sech–tanh' method, the 'harmonic balance' method and so on, should be mentioned. In these methods, the solution of

a nonlinear PDE is represented by a combination of trigonometric or hyperbolic functions with indefinite coefficients. These coefficients are determined by substituting this form of solution into the equation. In general, such methods are of no interest in the context of the soliton theory. The maximum of what can be defined using these methods is some hump-like solutions. These solutions, applying such methods to a nonlinear PDE, are usually declared to be solitons. However, the hump-like structures cannot be termed 'solitons' unless solutions describing their interactions as solitons are constructed.

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
