# Peer review of "Lax Pairs for the Modified KdV Equation"

_axioms, doi:10.3390/axioms13020121_

Round 1

Reviewer 1 Report

Comments and Suggestions for Authors

See attached

Author Response

I am grateful to the Reviewer for encouraging comments.

Reviewer 2 Report

Comments and Suggestions for Authors

In this manuscript, we remark that the present study establishes multi-parameter families of Lax pairs for the modified Korteweg-de Vries (mKdV) equation using a novel direct method. This approach not only defines the Lax pairs but also identifies gauge transformations that convert them into simpler forms. Additionally, we explore the broader implications of the direct method, considering its potential applications to various types of evolution equations. The study delves into the intricacies of the gauge transformations, shedding light on their significance in simplifying the mathematical representation of the mKdV equation and potentially extending their utility to other related equations. Furthermore, we discuss the versatility of the direct method, paving the way for a deeper understanding of its applicability beyond the confines of the mKdV equation.

The paper, therefore, is interesting and valid. However, a number of changes should be incorporated to improve the text. The required improvements are listed below.

1- In order to enable readers to understand the content of the paper easily, the manuscript must be carefully refined and checked. Indeed, there are many misplaced punctuation in the manuscript, also lacks of commas, and points in several places, the spacing between words is not respected.

2- The introduction should provide an overview of the various approaches employed in solving partial differential equations.

3- Recent works about PDEs should be cited in the introduction such as

Solution of an ice melting problem using a fixed domain method with a moving boundary.   Bull. Math. Soc. Sci.Math. Roum. 2019, 62, 341–353. I would like to add this reference which is more recent than you have cited.

4- The organization of the paper should be reviewed to be consistent with the
rest of the paper.

Comments on the Quality of English Language

 I think that the english write of the manuscript should be revise, and also all typographic errors in the text should be corrected.

Author Response

I am grateful to the Reviewer for such a careful reading of my paper and suggestions that led to improvement of the presentation. 

The following has been made to address the Reviewer comments.

(1) The manuscript has been carefully checked, missprints have been corrected. 

(2) An overview of the methods of solving nonlinear partial differential equations has been included. 

(3) Several references to recent works on PDEs have been added. The reference suggested by the Reviewer has been added. 

(4) The paper has been partially reorganized. 

Thus, all Reviewer's suggestions have been incorporated.      

Reviewer 3 Report

Comments and Suggestions for Authors

In my opinion, this is a preliminary theoretical study on application of the direct method to the   modified Korteweg–de Vries (mKdV) equation. The author demonstrated that this method can produce multiparameter families of the Lax pairs. 

I consider that the importance of the subject was well highlighted in the paper, in accordance with the relevance of the approach for this type of equations.

I suggest the author develop future articles with applications. Therefore, I consider that the paper has the necessary quality for publication, in its current version, and I do not see any other comments about the paper.

Author Response

I am grateful to the Reviewer for encouraging comments and suggestion to develop future articles with applications.